# Copula-Based Joint Drought Index Using Precipitation, NDVI, and Runoff and Its Application in the Yangtze River Basin, China

**Hongfei Wei, Xiuguo Liu** , **Weihua Hua, Wei Zhang \*** , **Chenjia Ji and Songjie Han**

School of Geography and Information Engineering, China University of Geosciences (Wuhan),
Wuhan 430074, China; weihongfei@cug.edu.cn (H.W.); liuxiuguo@cug.edu.cn (X.L.);
huaweihua@cug.edu.cn (W.H.); jcj17@cug.edu.cn (C.J.); sondrahan@cug.edu.cn (S.H.)
\* Correspondence: weizhang@cug.edu.cn

**Abstract:** Drought monitoring ensures the Yangtze River Basin's social economy and agricultural production. Developing a comprehensive index with high monitoring precision is essential to enhance the accuracy of drought management strategies. This study proposes the standardized comprehensive drought index (SCDI) using a novel approach that utilizes the joint distribution of C-vine copula to effectively combine three critical drought factors: precipitation, NDVI, and runoff. The study analyzes the reliability and effectiveness of the SCDI in detecting drought events through quantitative indicators and assesses its applicability in the Yangtze River Basin. The findings are as follows: (1) The SCDI is a highly reliable and applicable drought index. Compared to traditional indices like the SPI, VCI, and SRI, it has a consistency rate of over 67% and can detect drought events in more sensitive months by over 51%. It has a low false negative rate of only 2% and a false positive rate of 0%, making it highly accurate. The SCDI is also applicable to all the third-level sub-basins of the Yangtze River Basin, making it a valuable tool for regional drought monitoring. (2) The time lag effect of the NDVI can affect the sensitivity of the SCDI. When the NDVI time series data are shifted forward by one month, the sensitivity of the SCDI in detecting agricultural drought improves from 47.8% to 53%. (3) The SDCI can assist in monitoring drought patterns in the Yangtze River Basin. From 2001 to 2018, the basin saw fluctuations in drought intensity, with the worst in December 2008. The western region had less frequent but more intense and prolonged droughts, while the eastern part had more frequent yet less severe droughts.

**Keywords:** drought index; C-vine copula; precipitation; NDVI; runoff



## 1. Introduction

As a complex natural disaster, drought profoundly impacts the ecological environment and social and economic development [1]. Against the backdrop of global warming, a noticeable rise in the frequency of droughts has been observed. There is the anticipation that upcoming droughts will exhibit greater severity and a wider extent [2].

Drought can be categorized into four types: meteorological, hydrological, agricultural, and socioeconomic drought [3]. The drought index is an essential means to monitor drought [4]. In order to reflect different types of droughts, domestic and foreign scholars have constructed hundreds of drought indices [5]. The meteorological drought index is linked to variables such as precipitation and evapotranspiration, and it primarily describes water scarcity resulting from insufficient precipitation [6], such as SPI. The agricultural drought index is linked to variables such as vegetation coverage area [7]. Its primary function is to describe a situation in which vegetation cannot grow adequately because of insufficient soil moisture. Standard indices in this category include the Standardized Soil Moisture Index (SSI) [8] and the Soil Moisture Anomaly Percentage Index (SMAPI) [9]. The hydrological drought index is connected to variables such as runoff and groundwater levels [10]. It describes the reduction of water volume in rivers or reservoirs due to

insufficient surface or groundwater, and examples of this type of index are the SWSI [11] and SRI [12]. The above are single-element drought indices, which can only capture specific phases of drought and may not reflect the overall trend or internal mechanisms of the process [13–15].

In recent years, a growing trend has emerged in developing comprehensive drought indices that integrate multiple sources of information, facilitating more precise monitoring and assessment of drought conditions [16,17]. Such indices aim to capture the complexities of drought by incorporating various relevant variables. For instance, Zhang et al. [18] employed a linear combination method to create the Microwave Integrated Drought Index (MIDI). This index has demonstrated effectiveness in short-term drought monitoring. Similarly, Chang et al. [19] constructed the comprehensive drought index PRSM, utilizing rainfall, soil water content, and runoff information through principal component analysis. However, it is essential to acknowledge that the commonly used linear combination method and principal component analysis may involve some degree of subjectivity in selecting and combining variables, leading to potential errors and affecting the reliability of drought monitoring results [20]. To address these concerns, some researchers have turned to the copula function, which can account for different marginal distributions of variables and avoid information loss during the conversion process. For example, Li et al. [21] employed the empirical copula method to calculate the observed joint score of SPEI and SSI, proposing the Modified Multivariate Standardized Drought Index (MMSDI) to enhance the effectiveness of meteorological and agricultural comprehensive drought monitoring.

Current drought indices require improvements because of the following factors. Firstly, drought assessments should take into account the entire water cycle, including meteorological, hydrological, and agricultural factors [22]. More than classical binary variables, like precipitation and NDVI (Normalized Digital Vegetation Index), are required [23]. Secondly, the mutual feedback and time lag between vegetation and dry conditions are essential to constructing the drought index [24]. However, this subject has yet to be thoroughly studied. Vegetation changes reflect the region's dry and wet conditions and the connection among soil, atmosphere, and water [25]. Therefore, it is necessary to include vegetation elements in the drought index and consider the hysteresis of vegetation in the water cycle [26]. Thirdly, most of the variable data used in the comprehensive drought index need to have the advantage of combining site observation data and remote sensing data. The accuracy of ground observation data is high, and remote sensing data can obtain a wide range of real-time vegetation conditions, soil moisture, and other data [27]. Therefore, combining ground station and remote sensing data has become necessary for future comprehensive drought index research [28].

Among historical drought events, the Yangtze River Basin is susceptible to summer high-temperature droughts from June to August [29], but their spatial distribution varies. In August, drought conditions typically affect the entire basin, while in September, they become more localized, primarily impacting certain areas in the middle and lower reaches [30]. Consequently, conducting drought analysis in the Yangtze River Basin at the seasonal or annual scale is inadequate for accurately pinpointing the specific areas affected by drought in each month [31]. In recent years, sudden drought events in the Yangtze River Basin have been on the rise [32]. Detecting these abrupt drought events accurately over extended time scales proves to be challenging [33]. Generally, the smaller the time scale, the more pronounced the changes in the drought index over time, which can provide a better reflection of monthly drought variation throughout the year [34]. Consequently, the monthly scale drought index is better suited for identifying short-term and sudden drought events, facilitating more accurate detection and aiding the government in making timely decisions. Drought assessments should select time scales in accordance with specific needs to enhance the effectiveness of drought evaluation [35]. The SPI of a one-month time scale can provide a better reflection of short-term meteorological drought [36]. Monthly vegetation coverage responds to precipitation with a lag effect of 1–2 months [37], while the SRI of a one-month time scale is more suitable for capturing monthly variations in hydrological drought [33].

Given these considerations, it is more practical to construct a comprehensive drought index suitable for the Yangtze River Basin from the perspective of a monthly scale.

The Yangtze River Basin is characterized by abundant precipitation, diverse vegetation types, and significant seasonal and interannual variations in runoff [38]. Drought in the Yangtze River Basin is a combination of meteorological, hydrological, and agricultural characteristics, which cannot be evaluated only using traditional single drought. Therefore, a comprehensive drought index was developed in this paper. Through our study, we successfully developed the Standardized Comprehensive Drought Index (SCDI), which effectively detects drought occurrences in the Yangtze River Basin. The SCDI utilizes a sophisticated multivariate copula joint distribution that integrates data on precipitation, NDVI (Normalized Difference Vegetation Index), and runoff for accurate and reliable results. Meanwhile, our study evaluated the effectiveness of the SCDI by comparing it to single-element drought indices in terms of similarities and differences. We also examined the rate of missing reports and false alarms. Furthermore, we analyzed drought's temporal and spatial variation characteristics in the Yangtze River Basin from 2001 to 2018, considering drought frequency, duration, and intensity factors. This research provides valuable insight into drought patterns in the Yangtze River Basin. The findings can provide valuable insight for making informed scientific decisions regarding future drought prevention and preparedness.

## 2. Materials and Methods

### 2.1. Study Area

The Yangtze River Basin lies between 24°30′N~35°45′N and 90°33′E~122°25′E, covering an overall expanse of roughly 1.8 million square kilometers. The basin spans 19 provinces and extends into the downstream plain, ultimately flowing into the East China Sea. The basin receives an average yearly precipitation of 1036 mm, with a general decrease in precipitation from southeast to northwest. The basin's topography is complex, mainly consisting of plateaus, mountains, hills, plains, and basins. The annual average temperature is 13.3 °C, showing a characteristic of higher temperatures in the southeast and lower temperatures in the northwest. Many tributaries and lakes are located in the basin, and precipitation and snowmelt have formed rich water resources, resulting in high runoff.

Over the last few decades, the Yangtze River Basin has encountered numerous extreme drought events as a consequence of both changes in climate patterns and human interventions [39,40]. Monitoring drought events is crucial in rationally allocating water resources in the Yangtze River Basin [41]. Traditional drought investigations often segment drought into different types, leading to insufficiency in comprehensive drought assessment. Given the extensive expanse, intricate topography, and diverse populace, economy, and natural attributes of the Yangtze River Basin, the conventional drought index must precisely discern drought occurrences. Therefore, this paper divided the whole basin into 45 third-level sub-basins (Figure 1), constructed the drought index for the 0th sub-basin, verified the universality of the index in all sub-basins, and then evaluated and analyzed the overall drought in the Yangtze River Basin.

### 2.2. Data

Monthly precipitation data were acquired from the China Meteorological Data Service Center (http://data.cma.cn/, accessed on 30 June 2023), which provides monthly scale precipitation data from national-level surface monitoring stations. The temporal resolution was one month, and the spatial resolution was site scale.

The NDVI was obtained from the Resource and Environment Science and Data Center (https://www.resdc.cn/, accessed on 30 June 2023). The dataset was derived from a continuous time series of SPOT/VEGETATION satellite remote sensing data. The temporal resolution was one month, and the spatial resolution was 1 km.

The dataset of natural runoff (CNRD v1.0) was obtained from the National Tibetan Plateau Data Center (https://data.tpdc.ac.cn/, accessed on 30 June 2023) [42]. Compared to the global runoff grid datasets ISIMIP and GRUN, this dataset exhibits a more continuous

transition in the spatial distribution of runoff [43], making it more suitable for complex terrain and various climate models in China. The temporal resolution was one month, and the spatial resolution was $0.25° \times 0.25°$.

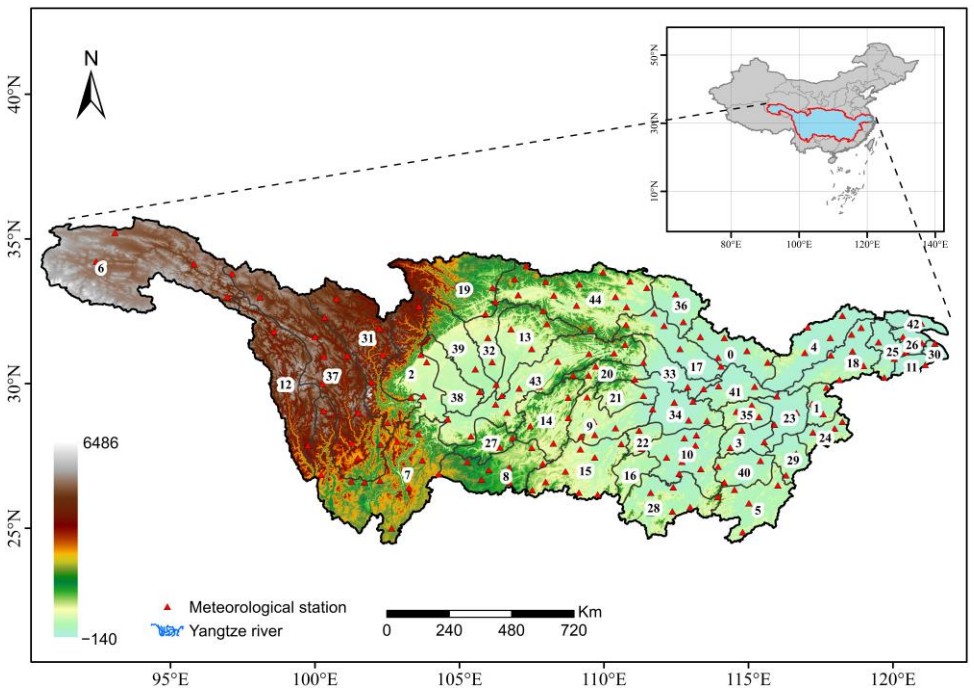

**Figure 1.** Study area.

The time range of the above precipitation, NDVI, and natural runoff dataset was from June 2001 to December 2018, with a total of 211 months. In order to unify the spatial resolution, this paper used ArcGIS software to obtain the NDVI and runoff time series data at the site locations so that the spatial resolutions of the precipitation, NDVI, and runoff were all set to the site scale.

## 2.3. Assumptions and Methodology

Figure 2 illustrates the procedural steps in constructing the SCDI, encompassing six key stages. Initially, one specific dataset was selected for constructing the SCDI in meteorology, agriculture, and hydrology. This study's selected datasets were precipitation, NDVI, and runoff data. Subsequently, the C-vine copula function established interdependencies among the three elements. The construction of a multivariate copula function is mainly divided into the following steps: selecting the appropriate marginal probability distribution and choosing the proper joint probability distribution. The marginal distributions of the three elements are fitted using the normal, gamma, log-normal, Weibull, and gen extreme value (GEV) distributions. The Akaike information criterion (AIC) is utilized to ascertain the most fitting distribution choices. A smaller AIC value indicates a more suitable theoretical distribution. Furthermore, potential pair-copulas for the two-dimensional copula, including the Gaussian, Student, Clayton, and Clayton's rotation structure copula, are chosen. The AIC, Bayesian information criterion (BIC), and log-likelihood (LL) determined the optimal pair-copula structure. A lower value for the AIC and BIC and a higher value for the LL indicate that the pair-copula distribution is the most suitable. The C-vine copula model was constructed, which integrates the meteorological, agricultural, and hydrological components. After building the model, the cumulative joint probabilities are standardized using the standard normal distribution transformation to receive the SCDI. Ultimately, the drought grade of the SCDI is classified based on a predefined threshold.

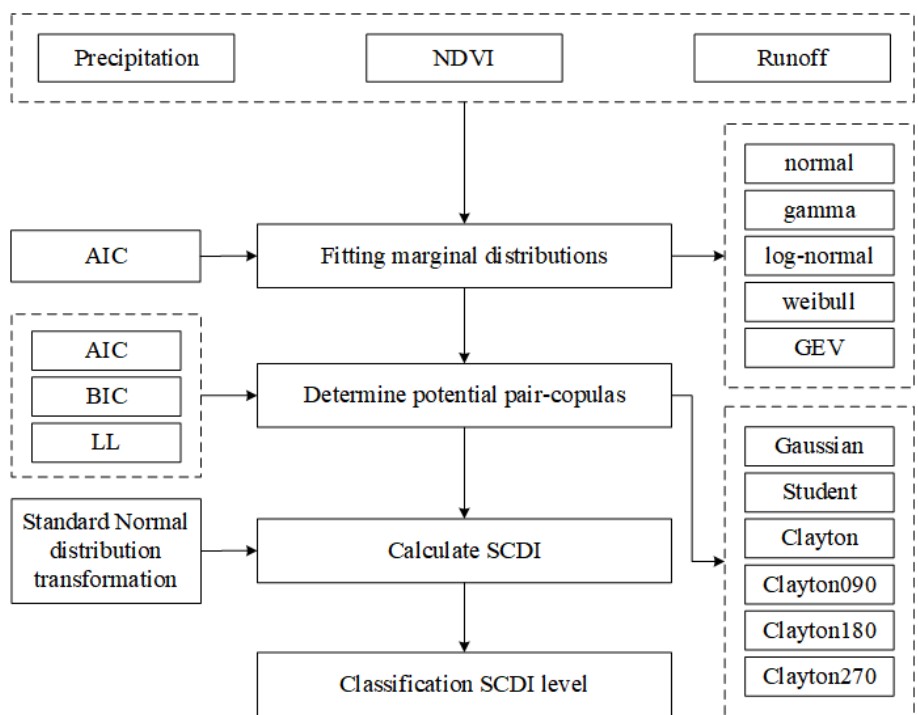

**Figure 2.** Diagram for Construction of the SCDI.

2.3.1. Single-Element Drought Indices Calculation

The standardized Precipitation Index (SPI) is a drought index that quantitatively characterizes regional drought in different time scales [44,45] and has the characteristics of simplicity in calculation and high accuracy. The SPI assumes that the frequency distribution of the precipitation follows the skewed distribution. The corresponding cumulative probabilities can be calculated through the skew distribution. Subsequently, the cumulative probabilities underwent a transformation to conform to a standard normal distribution, resulting in the calculation of the SPI [46]. Please refer to the referenced literature for specific SPI calculation methods and steps [47]. In this study, the SPI was at a monthly scale, where its value represents the degree of wetness or dryness. A higher value signifies a more humid condition, whereas a lower value signifies a drier situation. The precipitation data used to calculate SPI in this paper are monthly station precipitation data, and the monthly SPI time series was calculated using Python programming. Table 1 presents the drought severity classification of the SPI based on the China standard "Meteorological Drought Classification GB/T20481-2006" [48].

**Table 1.** Classification of drought grade.

| Drought Grade | Value of SPI | Value of SRI | Value of SCDI |
|---|---|---|---|
| Neutral | $SPI \geq -0.5$ | $SRI \geq -0.5$ | $SCDI \geq -0.5$ |
| Slight | $-1.0 < SPI \leq -0.5$ | $-1.0 < SRI \leq -0.5$ | $-1.0 < SCDI \leq -0.5$ |
| Moderate | $-1.5 < SPI \leq -1.0$ | $-1.5 < SRI \leq -1.0$ | $-1.5 < SCDI \leq -1.0$ |
| Severe | $-2.0 < SPI \leq -1.5$ | $-2.0 < SRI \leq -1.5$ | $-2.0 < SCDI \leq -1.5$ |
| Extreme | $SPI \leq -2.0$ | $SRI \leq -2.0$ | $SCDI \leq -2.0$ |

The NDVI was introduced by Rouse et al. [49]. It is related to vegetation coverage and primarily reflects its physiological and ecological characteristics. The NDVI is one of the most extensively employed remote sensing vegetation indices, and the calculation is relatively simple. However, it can only be applied to uniform and flat land surfaces and can only reflect the influence of single factors on vegetation. Monitoring drought in nonuniform regions may encounter various issues [50]. In order to solve the above impacts,

Kogan et al. [51] proposed the Vegetation Condition Index (VCI). When drought occurs, the vegetation is under water stress, and the NDVI value of the vegetation decreases to varying degrees. Therefore, the VCI can be used to reflect the soil moisture status [52]. The calculation formula is as follows:

$$VCI = \frac{NDVI_i - NDVI_{min}}{NDVI_{max} - NDVI_{min}} \tag{1}$$

where $NDVI_i$ is the value of the NDVI for the i-th period of a given year, $NDVI_{max}$ is the maximum value of the NDVI for the i-th period over multiple years, and $NDVI_{min}$ is the minimum value of the NDVI for the i-th period over numerous years. When the value of the VCI is between 0 and 0.3, this indicates poor vegetation growth and severe drought; between 0.3 and 0.7 indicates moderate vegetation growth and mild drought, and between 0.7 and 1 indicates good vegetation growth and no drought.

The Standardized Runoff Index (SRI) was proposed by Shukla et al. Equal to the concept of the SPI reference in 2008 [53], which can reflect the probability of runoff occurrence, the SRI utilizes the gamma distribution to describe the variation in the runoff. It involves normalizing the runoff data that follows a gamma distribution to obtain the SRI sequence. For the specific calculation methods and steps of the SRI, please refer to the referenced literature [54]. Similar to the SPI, the SRI is typically divided into five levels, as shown in Table 1.

### 2.3.2. Definition and Calculation of SCDI

The copula function was initially introduced by Sklar [55], in 1995, and was rigorously defined after a series of studies. If the joint distribution of N-dimensional random variables $X_1, X_2, X_3, \ldots, X_N$ is denoted as $H(x_1, x_2, \ldots, x_n)$, when the corresponding marginal distributions of the variables are $F(x_1), F(x_2), \ldots, F(x_n)$, then there exists an N-dimensional copula function $C(u_1, u_2, \ldots, u_n)$ such that the following equation holds:

$$H(x_1, x_2, \ldots, x_n) = C(F(x_1), F(x_2), \ldots, F(x_n)) \tag{2}$$

Function $C$ can be derived as follows:

$$C(u_1, u_2, \ldots, u_n) = H\left(F_1^{-1}(u_1), F_2^{-1}(u_2), \ldots, F_n^{-1}(u_n)\right) \tag{3}$$

The function $C$ exhibits inherent invariance, enabling a better capture of the correlation structure among random elements. This makes it superior to the Pearson correlation coefficient in accurately capturing the nonlinear correlations among the variables.

The expression of the joint probability density function $f$ is as follows:

$$f(x_1, x_2, \ldots, x_n) = C(F_1(x_1), F_2(x_2), \ldots, F_n(x_n)) \prod_{n=1}^{n} f_n(x_n) \tag{4}$$

This study chose theoretical probability distributions commonly used in hydrometeorology, including the normal, gamma, log-normal, Weibull, and GEV distributions. The parameters of each distribution were estimated by the maximum likelihood methodology [56], while AIC was employed to assess the fitting performance of each distribution function.

According to different classification criteria, multiple families of copula can be obtained. The most commonly used families in research are the Archimedean copula family [57] and elliptic copula family [58], which primarily comprises Frank, Clayton, Gumbel copula, and other types, and Gaussian and t copula. These two types of families are widely used because of their simple structure. However, because of the limitation of the parameters, they are mainly used to build two-dimensional joint distribution models, and they need to be more flexible for high-dimensional modeling. Vine copulas offer a means to tackle this concern. The subsequent explanation outlines the C-vine copula approach:

The n-dimensional density function is performed using the C-vine copula:

$$f(x_1, x_2, \ldots, x_n) \prod_{k=1}^{n} f(x_k) \prod_{j=1}^{n-1} \prod_{i=1}^{n-j} C_{j,j+i|1,\ldots,j-1} \left\{ F\left(x_j | x_1, \ldots, x_{j-1}\right), F\left(x_{i+j} | x_1, \ldots, x_{j-1}\right) \right\} \quad (5)$$

where $j$ is the different tree, and $i$ is the other root node.

C-vine copula is a copula construction method based on a tree-like structure. It decomposes the joint distribution of multidimensional random variables into a series of conditional distribution product forms through a stepwise modeling process. Its flexibility and interpretability make it suitable for modeling and analyzing high-dimensional data. The C-vine pair-copula structure is shown in Figure 3.

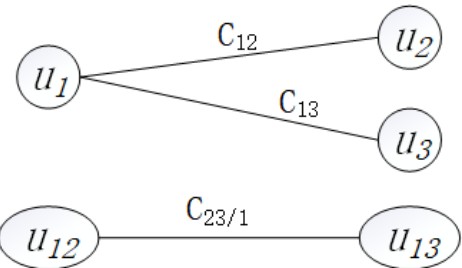

**Figure 3.** Decomposition structure of C-vine pair-copula with 3 variables.

According to the C-vine pair-copula structure above, the joint density function of the three-dimensional variables can be expressed as:

$$f(x_1, x_2, \ldots, x_n) = f(x_1) \cdot f(x_2) \cdot f(x_3) \cdot c_{12}(F(x_1), F(x_2)) \cdot c_{13}(F(x_1), F(x_3)) \cdot c_{23|1}(F(x_2|x_1), F(x_3|x_1)) \quad (6)$$

This study employed the widely used nested Archimedean construction within the Archimedean copula. The nested Archimedean building can be divided into fully nested Archimedean construction (FNAC) and partially nested Archimedean construction (PNAC). Unlike in FNAC, where the same Archimedean copula function was used at each layer, PNAC allows for different copula functions to be used in each layer. This provides more flexibility in constructing copula functions with varying correlation characteristics. Based on the advantages of PNAC, this study adopted the PNAC approach and selected Gaussian, Student, Clayton, and Clayton's rotated structure as potential pair-copulas. In assessing the developed C-vine copula model, AIC, BIC, and LL were employed to appraise each pair-copula and ascertain the optimal fit:

$$AIC = -2logL(\hat{\theta}) + 2m \quad (7)$$

$$BIC = -2logL(\hat{\theta}) + mlog(N) \quad (8)$$

where $m$ represents the count of the independent parameters, and $N$ signifies the sample size. *AIC* and *BIC* are deemed more favorable with lower calculated values, while *LL* is considered better with a higher estimated value. These aspects collectively indicate how much the model aligns with the data.

This study selected monthly scale precipitation, NDVI, and runoff sequences as random variables for constructing the joint distribution, creating a C-vine copula model. By employing the Kendall distribution transformation, the cumulative probability was mapped onto a single-dimensional plane. The joint probability can be delineated as follows:

$$P(U_1 < u_1, U_2 < u_2, U_3 < u_3) = \omega \quad (9)$$

where $u_1$, $u_2$, and $u_3$ are any value in the precipitation, NDVI, and runoff sequences, respectively.

According to Kao and Rao's method [59,60], the cumulative joint probability can be calculated as follows:

$$K(\omega) = P\left[C_{U_1,U_2,U_3}(U_1, U_2, U_3) \leq \omega\right] = q \tag{10}$$

In order to enable the comprehensive SCDI to perform a comparative analysis with the SPI, VCI, and SRI, the obtained cumulative joint probability needs to be standardized. The final expression of the *SCDI* is as follows:

$$SCDI = \varphi^{-1}(q) \tag{11}$$

where $\varphi$ represents the standard normal distribution function.

The SCDI combines the scalar values of the SPI, VCI, and SRI, which can collectively represent meteorological, agricultural, and hydrological drought scenarios. In this paper, according to the SPI and SRI drought classification methods, the drought is divided into five different levels: neutral, slight, moderate, severe, and extreme. The drought classification of the SCDI is shown in Table 1.

2.3.3. Evaluation Criterion of Index

In this study, the threshold for drought occurrence was set at $-0.5$ for the SPI, SRI, and SCDI, indicating drought occurrence when the value fell below $-0.5$. For the VCI, the threshold for drought occurrence was set at 0.5. To facilitate the comparative observation of the four index change curves, the SPI, VCI, SRI, and SCDI sequences were normalized using the zero-mean normalization function, where the drought threshold for each index was set to 0, and values less than 0 indicated drought occurrence. The formula for the zero-mean normalization function is as follows:

$$Z = \frac{x - \overline{x}}{\sigma} \tag{12}$$

$$\sigma = \sqrt{\frac{1}{N}\sum_{i=1}^{N}(x_i - \overline{x})^2} \tag{13}$$

where $x$ is the drought index sequence; $\overline{x}$ is the mean of the drought index sequence; $\sigma$ is the standard deviation of the data; $N$ is the length of the drought index series.

To quantitatively assess the efficacy of the composite index, the six aspects of correlation, consistency, sensitivity, accuracy, false positive rate, and false positive rate were used for the analysis. Pearson can describe the correlation between the SCDI and single-element drought index, and the formula is as follows:

$$R_{xy} = \frac{\sum_{i=1}^{n}[(X_i - E(X))(Y_i - E(Y))]}{\sqrt{\sum_{i=1}^{n}(X_i - E(X))^2 \sum_{i=1}^{n}(Y_i - E(Y))^2}} \tag{14}$$

where $X_i$ and $Y_i$ are the two datasets; $E(X)$ and $E(Y)$ are the mean values of the two datasets; and $n$ is the number of elements in the sequence.

Consistency can indicate whether the results of the drought characterization between the SCDI and single-element drought index were consistent. The higher the result, the better the ability of SCDI to characterize drought. The formula is as follows:

$$c = (a_i - TH_A)(b_i - TH_B) \tag{15}$$

$$con(A, B) = \frac{N}{S} * 100\% \tag{16}$$

where $a_i$ and $b_i$ are the values of index $A$ and index $B$ for the month $i$; $TH_A$ and $TH_B$ are the drought thresholds of index $A$ and index $B$; $c$ indicates whether index $A$ and index $B$

represent the exact characterization; *N* is the frequency at which *c* is greater than 0; and *S* is the total number of months in the index sequence.

The sensitivity can indicate the sensitivity of the SCDI in the drought characterization relative to the univariate drought index. The higher the result, the better the ability of the SCDI in the drought's characterization. The formula is as follows when index *A* and index *B* have represented the exact characterization:

$$H = |a_i| - |b_i| \tag{17}$$

$$sen(A, B) = \frac{N}{S} * 100\% \tag{18}$$

where $a_i$ and $b_i$ are the values of index *A* and *B* for the month *i*; *N* is the frequency at which *H* is greater than 0; and *S* is the total number of months.

The sensitivity is an important indicator that characterizes the comprehensive the drought index. Its results represent the proportion of months where the absolute value of the SCDI is more significant than that of the single-element drought index. More than 50% indicates a higher sensitivity of the SCDI and a better index performance.

### 2.3.4. Characteristics of Drought

Run-length theory is a common method for analyzing time series, which is widely used in the identification of drought events [61]. Herbst et al. [62] first used monthly precipitation data to identify drought with run theory. Utilizing run-length theory, the duration of drought (D) was determined with the extent of the negative run-length, and the drought intensity, S, was calculated using the area of the drought duration and section. In the identification of drought processes, it is necessary to filter small drought events and merge multiple drought events. The specific criteria for identifying drought events are as follows:

1.  If the SCDI equals 1 (i.e., SCDI = 1), it is directly considered a drought process.
2.  When D equals 1 (i.e., D = 1), if the monthly SCDI is less than −1.5, it is classified as a drought process; otherwise, it is not established as a drought event.
3.  If the time interval between two adjacent drought processes is only one month and the value of SCDI is below −0.5 for that month, they are considered a single continuous drought event. The total combined duration (D) is the sum of the duration of the two events plus 1, and the combined intensity (S) is the sum of their intensities.

## 3. Results

### 3.1. Joint Distribution of Three Elements

In this study, the joint distribution established was based on the third-level river basins of the Yangtze River Basin, aimed at constructing a comprehensive drought index of the third-level river basins. Therefore, for this part, the Wuhan-Hukou basin was chosen as the research object, corresponding to the sub-basin with the serial number 0 in Figure 1.

### 3.1.1. Selection of Marginal Distributions

Different marginal distributions were used to fit the long-term precipitation, NDVI, and runoff data separately, and the fitting results were obtained (Table 2). It can be observed that in the Wuhan-Hukou Basin, the GAMA distribution can better fit the precipitation and NDVI data. In contrast, the GEV distribution can better fit the runoff data.

**Table 2.** AIC results for each marginal distribution in the Wuhan-Hukou Basin.

| Distribution | Normal | Gamma | Log-Normal | Weibull | GEV |
|---|---|---|---|---|---|
| Precipitation | 3433 | 3311 | 3332 | 3312 | 3328 |
| NDVI | −260 | −268 | −265 | −265 | −266 |
| Runoff | 2232 | 2109 | 2081 | 2145 | 2074 |

### 3.1.2. Selection of Pair-Copula Distributions

The Gaussian, Student, Clayton, and Clayton's rotation structures, commonly used in hydrology [10], were used as alternative types of pair-copulas. The optimal pair-copula type for the Wuhan-Hukou Basin was determined based on the AIC, BIC, and LL.

In Table 3, the 1, 2, 3 in $C_{12}$, $C_{13}$ and $C_{23/1}$ represent the Wuhan-Hukou Basin's precipitation, NDVI, and runoff, respectively. The Gaussian distribution was the optimal pair-copula C12 for rainfall and NDVI according to the different criteria. The optimal pair-copula C13 for precipitation and runoff was the Clayton180 distribution. Under precipitation conditions, the Gaussian distribution was also the optimal pair-copula C23/1 for NDVI and runoff.

**Table 3.** Evaluation criteria for each pair-copula distribution in the Wuhan-Hukou Basin.

| Distribution | Norm. | Gaussian | Student | Clayton | Clayton090 | Clayton180 | Clayton270 |
|---|---|---|---|---|---|---|---|
| | AIC | **−91.66** | −16.66 | −56.86 | 2.18 | −85.67 | 2.21 |
| $C_{12}$ | BIC | **−88.31** | −9.96 | −53.51 | 5.53 | −82.32 | 5.56 |
| | LL | **46.83** | 10.33 | 29.43 | −0.09 | 43.83 | −0.1 |
| | AIC | −354.68 | −158.34 | −191.64 | 2.26 | **−423.03** | 2.27 |
| $C_{13}$ | BIC | −351.33 | −151.63 | −188.29 | 5.61 | **−419.68** | 5.62 |
| | LL | 178.34 | 81.17 | 96.82 | −0.13 | **212.51** | −0.13 |
| | AIC | **−16.31** | −0.43 | −9.39 | 2.09 | −12.52 | 2.09 |
| $C_{23/1}$ | BIC | **−12.96** | 6.28 | −6.05 | 5.45 | −9.17 | 5.45 |
| | LL | **9.15** | 2.21 | 5.69 | −0.04 | 7.26 | −0.05 |

The bold entries in the table represent better fitting indicators.

### 3.1.3. Reduction of Lag Effect

Firstly, SPI, VCI, SRI, and SCDI were calculated based on the raw data separately. Then, the sensitivities of the SCDI compared to the single-element indices were calculated. The sensitivity of the SCDI compared to the SPI was 58.5%, compared to the VCI, which was 47.8%, and compared to the SRI, it was 53.1%. The results shows that the SCDI had lower sensitivity and was less effective than the VCI in the characterization of the agricultural drought. This may be due to the lower sensitivity caused by the lag effect of the NDVI.

The autocorrelation function (ACF) and partial autocorrelation function (PACF) [61] both displayed a correlation between the time series data and its lagged versions, in which the PACF represents the relationship between the current observed value and the subsequent observed values after removing the effects of the shorter lags. The ACF and PACF plots focused primarily on the correlation coefficients that exceed a certain threshold. Correlation coefficients that surpass the threshold may indicate significant lagged relationships within the time series data (Figure 4). In the figure, each column represents the ACF and PACF of the different variables, where the blue area on the horizontal axis represents the confidence interval. It can be observed that all variables exhibit a first-order lag. However, the NDVI has a more robust first-order lag than the other variables, with the correlation coefficient reaching 0.8. Therefore, the lag of 1 month for the NDVI has the most significant impact on the SCDI. To mitigate the lag effect of the NDVI, the time series data of the NDVI is shifted forward by one month when calculating the SCDI. The sensitivity of the revised SCDI, considering the lag effect reduction, compared to the VCI, is found to be 53%. This indicates that the SCDI with the reduced lag effect of the NDVI performed better, demonstrating improved effectiveness.

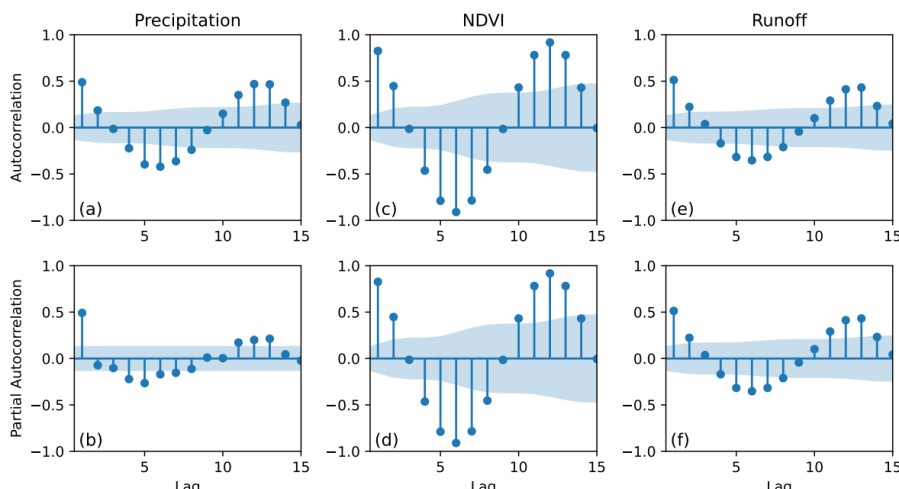

**Figure 4.** ACF and PACF for precipitation, NDVI, and runoff. (**a**,**b**) are the ACF and PACF results of precipitation respectively; (**c**,**d**) are the ACF and PACF results of precipitation respectively; (**e**,**f**) are the ACF and PACF results of precipitation respectively.

### 3.2. Performance of the SCDI

Combined with the modified index construction method, the SPI, VCI, SRI, and SCDI time series were calculated for the Wuhan-Hukou Basin monthly from June 2001 to December 2018. Each index was classified based on the drought grade classification. Figure 5a shows the results, with the SPI and SRI curves exhibiting relatively severe fluctuations, the VCI curve showing relatively stable fluctuations, and the overall pattern of the SCDI curve ranging between −2.5 and 2.5, demonstrating regular fluctuations.

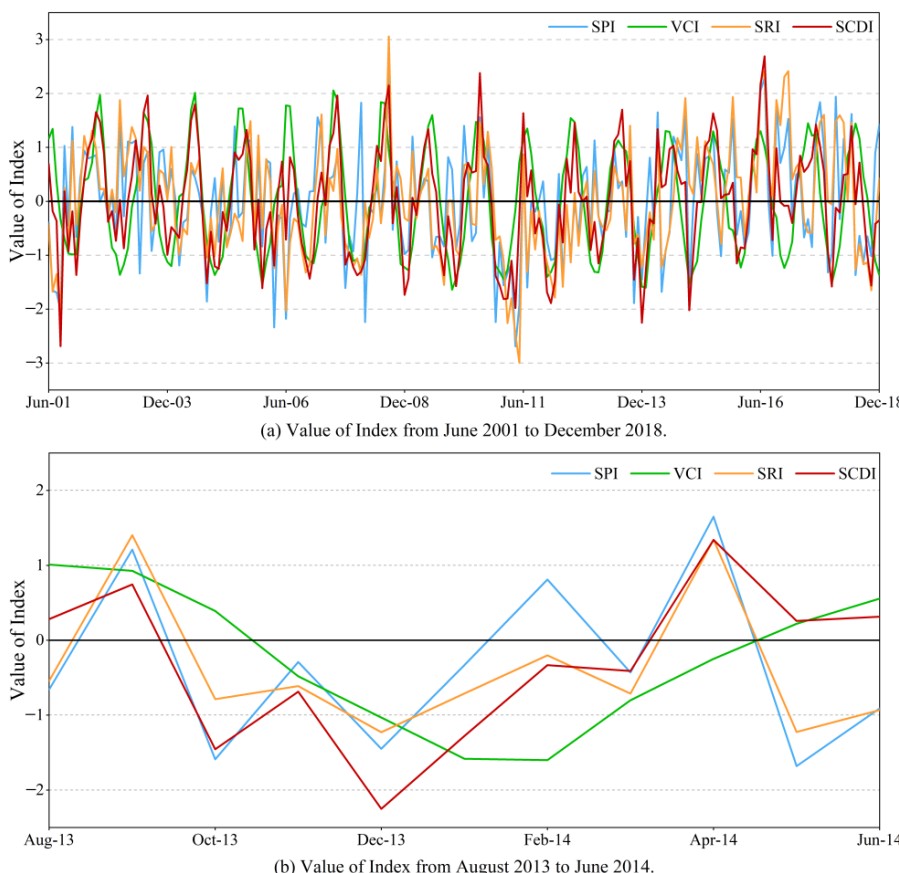

**Figure 5.** SPI, VCI, SRI, and SCDI of the Wuhan-Hukou Basin from June 2001 to December 2018.

3.2.1. Consistency, Sensitivity, and Accuracy

The Pearson correlation coefficients among the SCDI, SPI, VCI, and SRI were calculated. Table 4 shows that the correlation values among the SCDI and SPI, VCI, and SRI were above 0.6, indicating a high reliability of the SCDI based on the precipitation, NDVI, and runoff. Among the 211 monthly data samples analyzed, the SCDI was consistent with the SPI in 142 months, VCI in 176 months, and SRI in 145 months. The consistency of the SCDI with the SPI, VCI, and SRI was 67.3%, 83.41%, and 68.72%, respectively. The high consistency of the SCDI with each single-element drought index demonstrates its ability to characterize the drought characteristics in the Wuhan-Hukou Basin.

**Table 4.** Correlation, consistency, and sensitivity of the single-element indices with the SCDI.

| Norm. | SPI | VCI | SRI |
|---|---|---|---|
| Correlation | 0.60 | 0.76 | 0.61 |
| Consistency | 67.30% | 83.41% | 68.72% |
| Sensitivity | 60.15% | 51.10% | 53.10% |

In conjunction with the historical drought events observed from September 2013 to April 2014, as depicted in Figure 5b, it is evident that the SCDI detects the onset of drought earlier than the SPI and SRI. This indicates that the SCDI exhibits higher sensitivity in capturing the beginning of drought than meteorological and hydrological drought indices. Furthermore, the SCDI identifies the end of drought earlier than the SRI and VCI, demonstrating its superior capability in detecting the conclusion of drought compared to hydrological and agricultural drought indices. The SCDI integrates the characteristics of the SPI, VCI, and SRI. It not only captures meteorological drought based on precipitation but also identifies hydrological drought based on runoff. Additionally, it considers the lag effect of agricultural drought and provides accurate predictions of the end of the drought. As a result, the SCDI possesses distinct advantages in drought characterization.

The SCDI accurately captures the onset and end of drought and exhibits certain advantages in terms of sensitivity. The results indicate that the absolute value of the SCDI represents 59.15% of the total number of samples per month with consistent characterization, demonstrating its highest sensitivity. Although the sensitivity of the SCDI to VCI is 51.1%, which may seem relatively low, the consistency between the SCDI and VCI reaches a notable 83.41%. This high consistency is attributed to the correction of the NDVI hysteresis effect. Moreover, the SCDI demonstrates a better ability to characterize agricultural drought. Therefore, compared to single-element drought indices, the SCDI displays higher sensitivity in characterizing drought.

Based on the "China Meteorological Disaster Yearbook" and relevant literature [63], drought events occurred in the Wuhan-Hukou Basin during the following periods: September to November 2007, October 2010 to May 2011, September 2011 to April 2012, October to December 2012, November 2015 to April 2016, and October 2017 to February 2018. Figure 6 illustrates these drought periods, with the shaded areas in Figure 6a–f representing the corresponding drought events. The drought characteristics reflected by the SCDI align with the above occurrences, indicating the high accuracy of the SCDI in characterizing drought.

Additionally, Figure 7 illustrates the change curves of the SPI, VCI, SRI, and SCDI during two time periods: 2006–2007 and 2015–2016. In the drought event from September 2006 to February 2007 (Figure 7a), the SPI, representing meteorological drought, did not accurately capture the subsequent months of drought. The SRI, meaning hydrological drought, failed to capture the beginning and end of the drought accurately. The VCI, representing agricultural drought, exhibited a specific time lag effect. However, according to the analysis of high temperature and drought disasters by Fan Jinjin et al. [64], this drought event affected 8.982 million people in Hubei Province, with an affected area reaching approximately 1 million hectares. The drought caused the withering and death of surface plants, significantly impacting drinking water for humans and livestock, and

the SCDI accurately captured the severity of this drought event. In the drought event from November 2015 to March 2016 (Figure 7b), the SCDI, VCI, and SPI successfully captured the drought, while the SRI could not do so. This demonstrates that, compared to single-element drought indices, the SCDI provides a comprehensive assessment by considering multiple input variables, resulting in higher accuracy in capturing drought events.

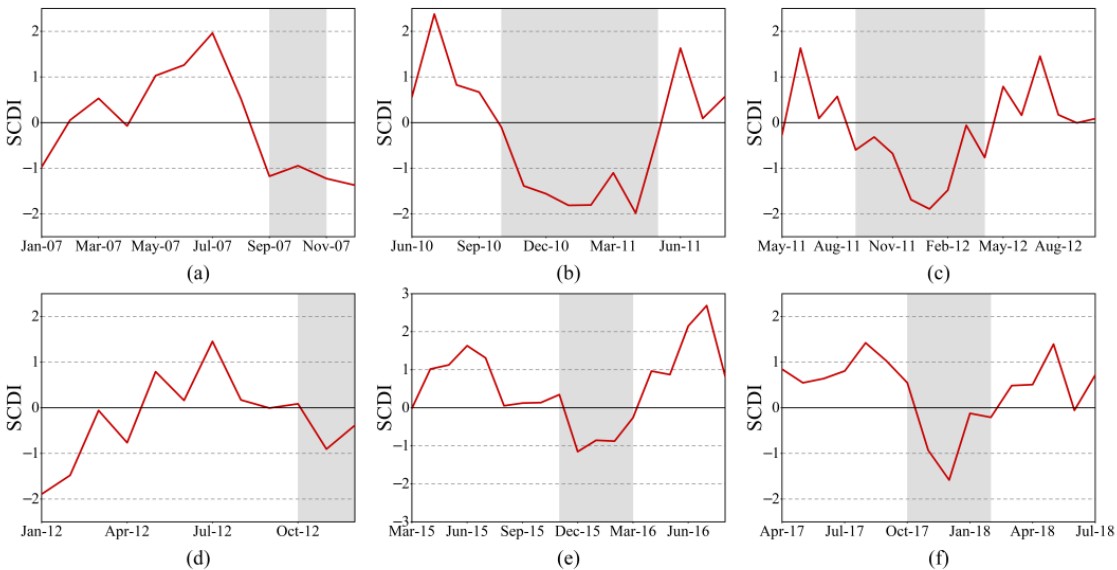

**Figure 6.** Performance of the SCDI in historical drought events. The gray part represents one drought event respectively. (**a**–**f**) are SCDI sequences from January 2007 to November 2007, from June 2010 to June 2011, from May 2011 to August 2012, from January 2012 to December 2012, from March 2015 to July 2016 and from April 2017 to July 2018 respectively. (**b**) shows the drought index sequence from June 2006 to September 2007.

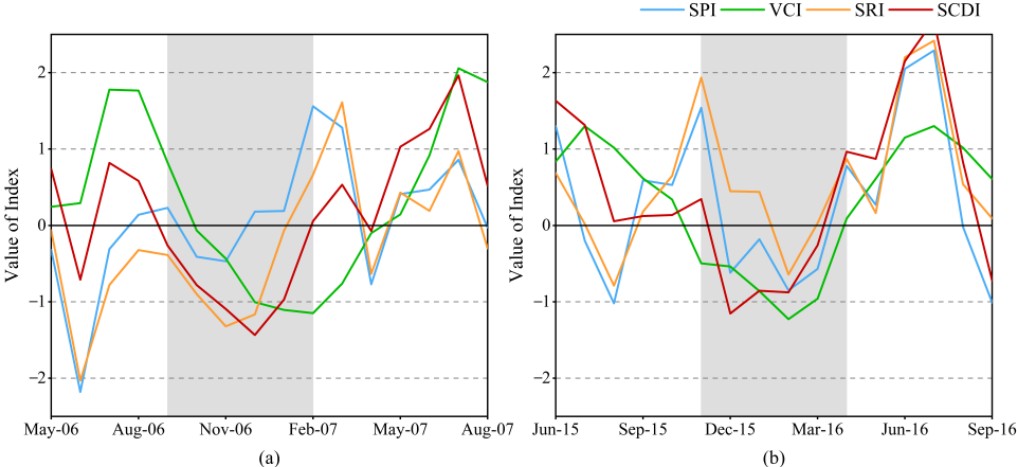

**Figure 7.** Performance of the SPI, VCI, SRI, and SCDI in historical drought events. The gray part represents one drought event respectively. (**a**) shows four drought index sequences from May 2006 to August 2007; (**b**) shows four drought index sequences from June 2015 to September 2016.

### 3.2.2. FNR and FPR

False positive rate (FPR) and false negative rate (FNR) are essential indicators for disaster prediction. However, this paper makes certain assumptions because of the need for monitoring data on long-term, daily-scale drought phenomena in the study area. When SPI, VCI, and SRI's characterization results indicate drought, the sample is classified as a drought event. Conversely, when the characterization results of the SPI, VCI, and SRI all indicate nondrought, the sample is considered a nondrought event. Underreporting

occurs when the SCDI characterizes a sample as drought while it is nondrought, while misreporting happens when the SCDI describes a sample as nondrought while it is drought. In this study, drought statistics based on the SCDI, SPI, VCI, and SRI are performed on 211 samples in the Wuhan-Hukou Basin. The FPR and FNR of the SCDI are presented in Table 5.

**Table 5.** Drought field statistics of the SPI, VCI, SRI and SCDI.

| Assessment Criteria | Representation | Times | Rate |
|---|---|---|---|
| False positive rate (FPR) | SPI > 0&VCI > 0&SRI > 0 | 49 | 2% |
| | SPI > 0&VCI > 0&SRI > 0&SCDI < 0 | 1 | |
| False negative rate (FNR) | SPI < 0&VCI < 0&SRI < 0 | 48 | 0 |
| | SPI < 0&VCI < 0&SRI < 0&SCDI < 0 | 0 | |

Table 4 shows that the false alarm and false negative rates of the SCDI in the Wuhan-Hukou Basin from June 2001 to December 2018 were low. This indicates that the SCDI possesses a solid capability to characterize drought conditions accurately.

### 3.2.3. Contrast between the SCDI and Single-Element Indices

Figure 8 illustrates the disparity (D) between the SCDI and the individual single-element drought indices. The scatter plot portrays this disparity in which the varying degrees of divergence are represented by different colors. Instances in which the SCDI aligns closely with the single-element indices, denoted by grey scatter points, indicate negligible distinctions ($-1.5 < D < 1.5$). On the other hand, blue scatter points indicate a significant negative disparity ($D < -1.5$), while yellow scatter points signify a significant positive disparity ($D > 1.5$). It is noteworthy that because of inherent dissimilarities in the construction methodologies of the SCDI and single-element drought indices, their alignment signifies comparable drought severity. This minor divergence has minimal implications for drought management decision-making processes. Conversely, when the disparity in drought severity between the two indices exceeds a certain threshold, typically one grade, this denotes distinct drought conditions. As a result, a threshold of $|1.5|$ was established to determine whether a meaningful difference exists between the SCDI and single-element drought indices.

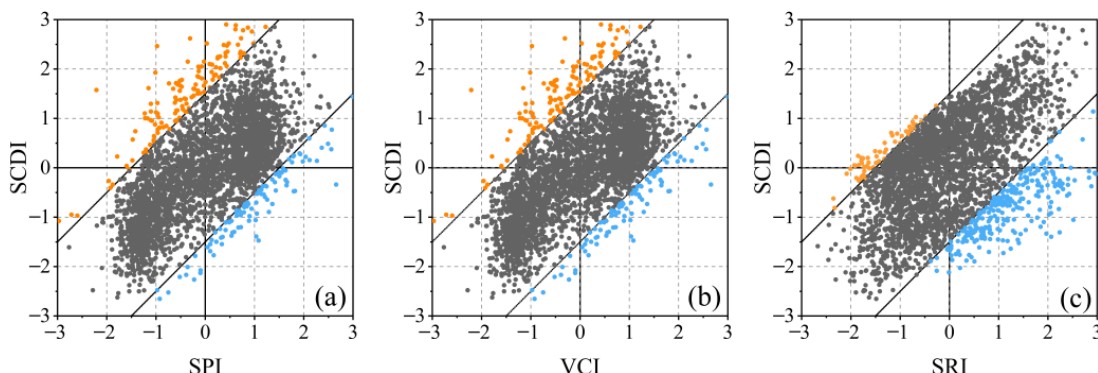

**Figure 8.** The differences between the SCDI and the SPI, VCI, and SRI in the Wuhan-Hukou Basin. (**a**–**c**) are scatter plots of differences between SCDI and SPI, VCI and SRI, respectively. Blue scatter plots represent significant negative differences and yellow scatter plots represent significant positive differences.

Figure 8a–c show that nearly all scatter points exhibit a 45° linear symmetry, indicating a similar overall trend between the SWCI and SPI and the VCI and SRI. Notably, the differences observed in the scatter points are predominantly cluster within the second and fourth quadrants. This indicates that the SCDI exhibits less variability in transitioning between wet and drought conditions compared to the individual single-element drought

indices. Specifically, the SCDI demonstrates reduced values in wet conditions in contrast to the individual single-element drought indices. Conversely, in instances of drought conditions, the SCDI indicates elevated values compared to the univariate indices. The primary reason for this behavior lies in the composition of the SCDI, which incorporates multiple elements of the water cycle, including precipitation, vegetation, and soil moisture. During droughts, the reduced precipitation disrupts the original water balance. To establish a new equilibrium, vegetation and soil supplement the water cycle by contributing their water. As a result, meteorological, agricultural, and hydrological droughts become more severe, leading to higher drought states for the SCDI compared to the SPI, VCI, and SRI. On the other hand, when new precipitation occurs, vegetation and soil absorb water to replenish the lost resources, thereby maintaining the water balance. Consequently, the SCDI exhibits lower values during wet conditions compared to the SPI, VCI, and SRI. Furthermore, it is worth noting that the soil's water absorption efficiency surpasses vegetation's, contributing to a more significant difference between the SCDI and the single-element drought indices in the fourth quadrant. A comprehensive analysis of the differences between the single-element drought indices and the SCDI reveals that while the former considers only single factors when characterizing drought, the latter considers multiple factors, resulting in a more accurate representation of the comprehensive drought situation.

### 3.3. Universality Analysis of the SCDI

In order to assess the applicability of the SCDI to all third-level sub-basins in the Yangtze River Basin, this study calculated the SCDI and single-element drought indices for all third-level sub-basins from June 2001 to December 2018 (excluding the 16th sub-basin because of the lack of a meteorological station). Figure 9 illustrates the correlation, consistency, and sensitivity between the SCDI and each single-element drought index in every sub-basin. Regarding correlation, most regions showed a correlation coefficient between the SCDI and the SPI, VCI, and SRI ranging from 0.5 to 0.8. However, some northwest parts displayed a low correlation coefficient with the SPI and SRI but a high correlation coefficient with the VCI. As for consistency, in most regions, the probability of the SCDI being consistent with the SPI and SRI ranged from 50% to 80%. Notably, the SCDI and VCI exhibited high consistency across the entire basin, particularly in the central and western regions, with a probability of consistent representation between 80% and 100%. Concerning sensitivity, the SCDI showed lower sensitivity (less than 50%) compared to the SPI and SRI for only one sub-basin each. The sub-basins with higher sensitivity than the SPI were mainly northeast of the Yangtze River Basin. In comparison, those with a higher sensitivity than the SRI were primarily located in the northwest. Compared with the VCI, the SCDI had four sub-watersheds with less than 50% sensitivity. Still, these four sub-watersheds performed well in the consistency of the SCDI and VCI, and the probability of characterization consistency was between 80% and 100%. Overall, the SCDI, based on precipitation, NDVI, and runoff data, demonstrated robust dependability when applied to the Yangtze River Basin. It effectively captured and characterized drought-related patterns within each of third-level sub-basins of the Yangtze River Basin.

Figure 10 illustrates the false negative and false positive rates of drought events captured by the SCDI from June 2001 to December 2018. In the figure, both the value of the FNR and FPR of all sub-basins fell within the range of 0–20%. The FNR surpassed 15% in the five sub-basins, with notable concentrations observed predominantly in the western and eastern sectors of the Yangtze River Basin. Only one sub-basin exhibited an FPR greater than 15%. Among the sub-basins, five exhibited FNR values exceeding 15%, with their primary concentration occurring within the western and eastern regions. Meanwhile, a value of FPR exceeding 15% was observed in only one sub-basin in the northern part of the Yangtze River Basin. In general, the SCDI displayed a solid ability to accurately characterize drought within the central region of the Yangtze River Basin. Still, its performance could be better in the western and eastern areas. Nonetheless, all values fell within a reasonable

range. As a result, the SCDI is well-suited for drought monitoring in all three sub-basins of the Yangtze River Basin, with an overall positive monitoring effect.

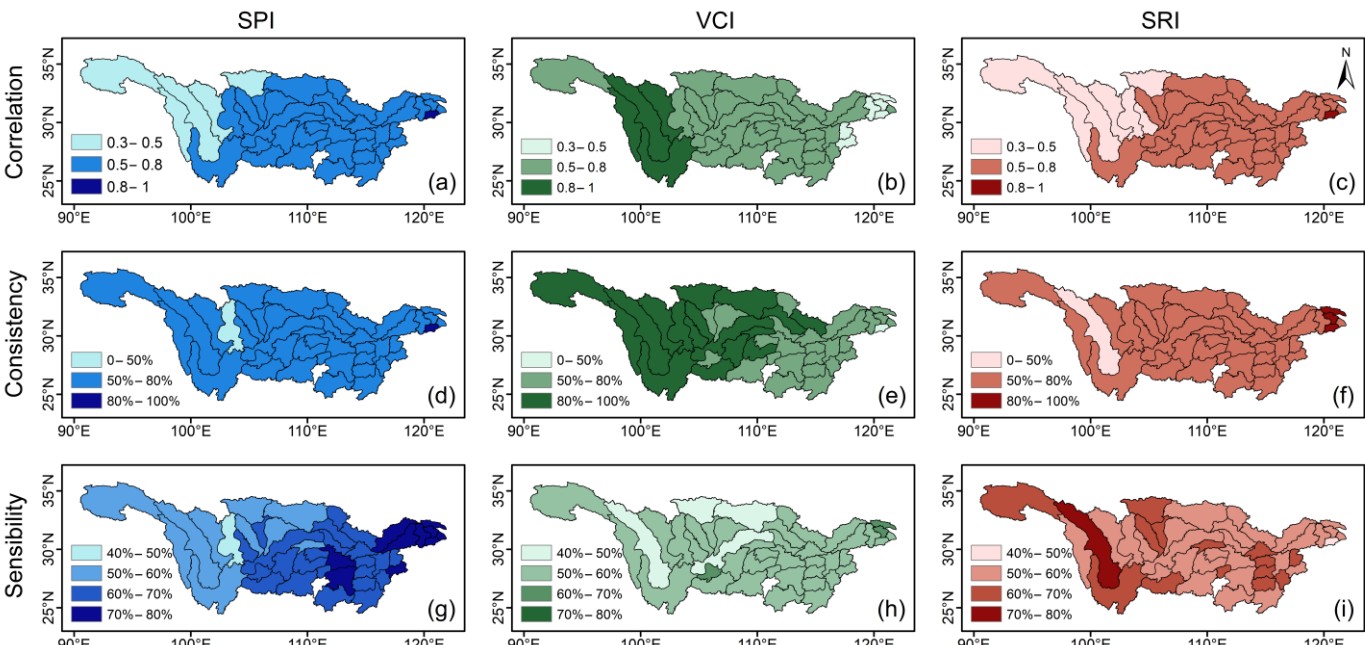

**Figure 9.** Correlation, consistency, and sensitivity of the SCDI with the SPI, VCI, and SRI in the third-level sub-basins of the Yangtze River Basin from June 2001 to December 2018: (**a**–**c**) correlation; (**d**–**f**) consistency; (**g**–**i**) sensitivity.

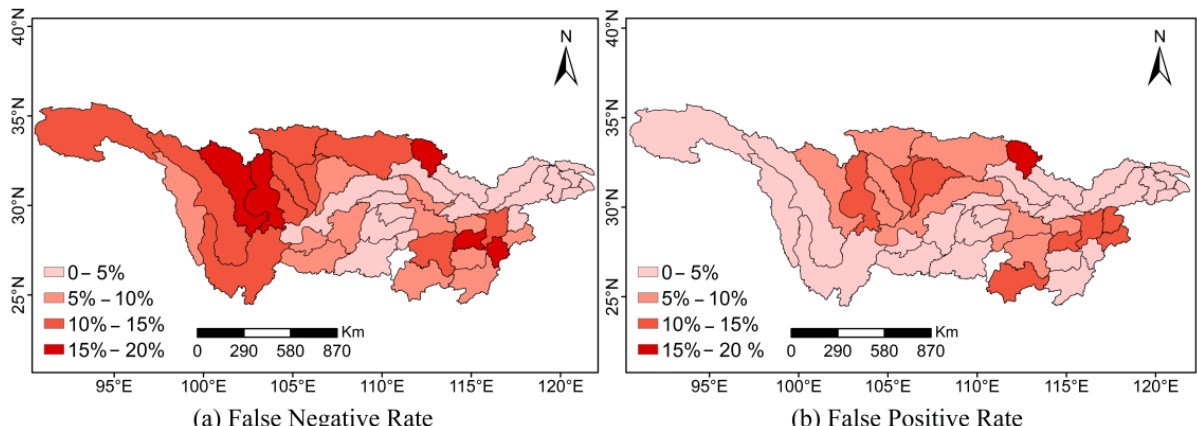

**Figure 10.** FNR and FPR of the SCDI in the third-level sub-basins of the Yangtze River Basin from June 2001 to December 2018.

### 3.4. Analysis of Drought Characteristic Evolution

3.4.1. Temporal Evolution Feature

The proportion of each drought grade in the Yangtze River Basin, as monitored by the SCDI from June 2001 to December 2018, fluctuated (Figure 11). From 2001 to 2010, the proportion of drought areas gradually increased, but from 2010 to 2018, it showed a gradual decrease, with a notable mutation occurring in December 2017. The years 2009, 2010, and 2011 experienced relatively severe drought, with proportions of 61.64%, 53.92%, and 67.03%, respectively.

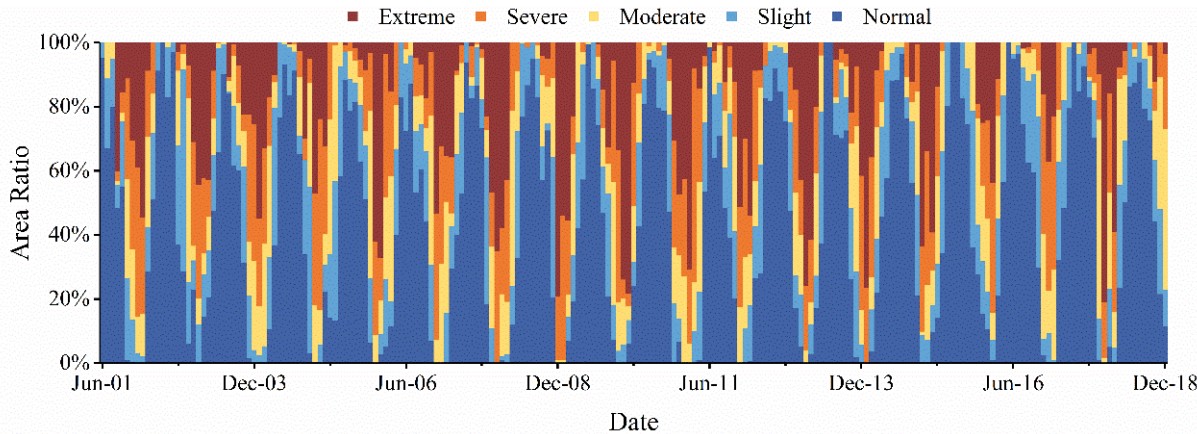

**Figure 11.** Time series of the area percentage for different drought classifications during 2001–2018 based on the SCDI in the Yangtze River Basin.

3.4.2. Spatial Evolution Feature

Figure 12 displays the spatial distribution of drought during the most severe drought month within the study period. This month reveals that the drought grade distribution in the Yangtze River Basin follows the spatial pattern of "severe–light–severe" from west to east, exhibiting apparent spatial heterogeneity. The regions both upstream and downstream of the Yangtze River exhibit regional severe drought, with the Qingyi-Minjian River Basin, Tangbai River Basin, and Qingge-Shuiyang River Basin experiencing extreme drought.

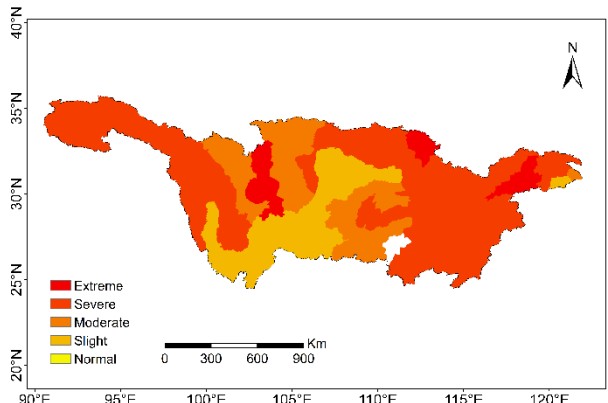

**Figure 12.** Spatial distribution of the SCDI in various sub-basins of the Yangtze River Basin in December 2008.

Figure 13a–c show that the occurrence rate of drought events in the Yangtze River Basin varied between 17 and 30 from 2001 to 2018, averaging 1 to 2 droughts yearly. The cumulative number of droughts in the regions both upstream and in the middle of the Yangtze River was less than 23, while only the northeastern part experienced close to 30 drought occurrences. The cumulative drought duration and intensity in the Yangtze River Basin gradually decreased from west to east. In most sub-basins, the cumulative drought duration did not exceed 74 months, and the power did not surpass 146. Only a few areas, such as the Yanglong and Tongtian Rivers, exhibited longer drought durations and higher intensities.

The frequency of the SCDI droughts of different grades at the monthly scale in each sub-basin of the Yangtze River Basin was calculated and presented in Figure 14. Based on the average frequency of each drought grade, it can be inferred that the overall drought situation in the Yangtze River Basin is as follows (from highest to lowest frequency): slight > extreme > moderate > severe. Notably, the sub-basins with serial numbers 6, 12, 31, and 37 exhibited more severe and extreme drought frequencies.

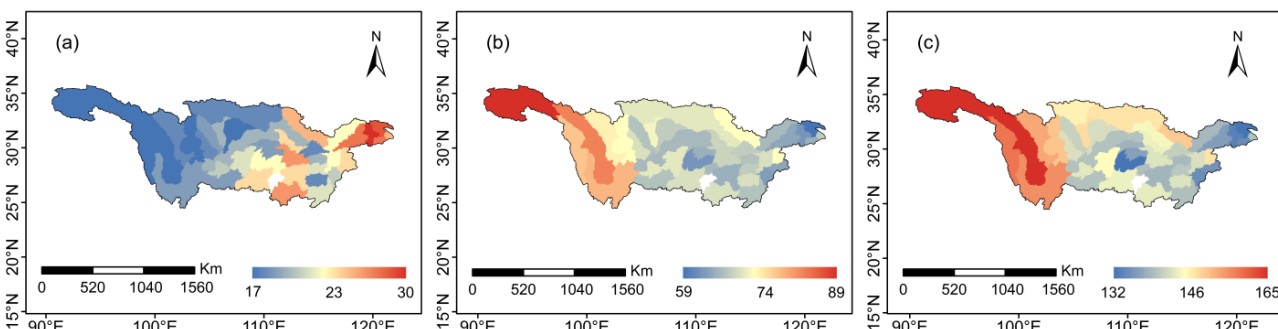

**Figure 13.** Spatial distribution of the comprehensive drought characteristics in the Yangtze River Basin from 2001 to 2018: (**a**) spatial distribution of drought frequency; (**b**) spatial distribution of drought duration; (**c**) spatial distribution of drought intensity.

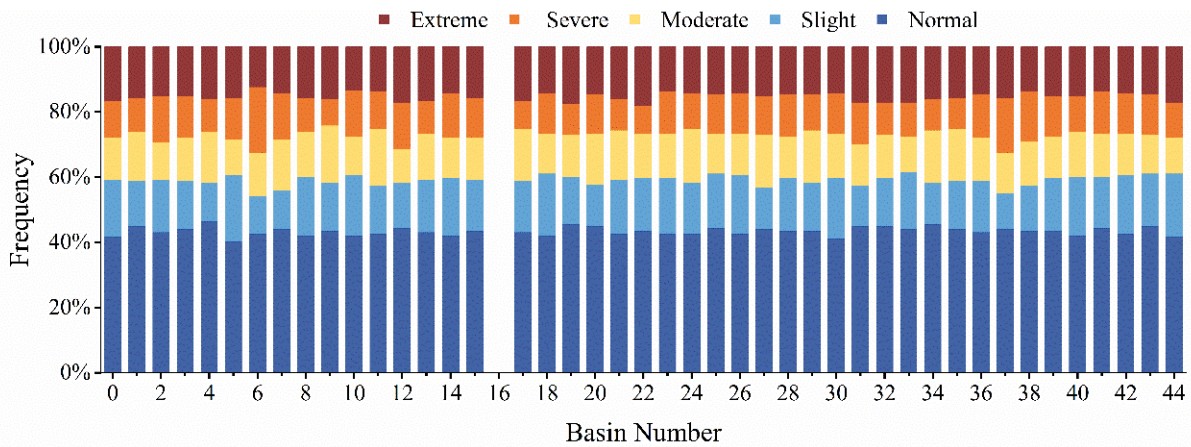

**Figure 14.** Drought frequency of the different levels in the monthly SCDI for various sub-basins in the Yangtze River Basin from 2001 to 2018.

## 4. Discussion

### 4.1. Selection of Multivariate Copula Parameters

The potential differences in the fitting distribution among various subwatersheds arise because of topographical and other factors. For instance, the Gaussian copula function has been found to provide a better simulation of drought characteristics in the Lake Poyang Basin [65], while the Frank Copula function is more suitable for analyzing drought frequency in the Nanpan River Basin [66]. Consequently, the optimal fitting distribution for the different subwatersheds within the Yangtze River Basin may vary. Given the extensive coverage of the Yangtze River Basin, it is not feasible to apply a single copula function across all three-tier basins. Therefore, it is essential to compute specific results for each sub-basin to determine the optimal marginal distribution and copula distribution.

According to the results in Table 2, the optimal marginal distributions for the Wuhan-Hukou Basin are GAMA and GEV. Likewise, the results in Table 3 indicate that the most optimal copula distributions for the Wuhan-Hukou Basin are Gaussian and Clayton180 distributions. By calculating the SCDI time series of 45 sub-basins in the Yangtze River Basin, it was found that there are differences in the optimal marginal distribution and copula distribution of all sub-basins. However, as depicted in Figures 9 and 10, the SCDI demonstrates significant improvements in correlation, sensitivity, consistency, false negative rates, and false positive rates across all sub-basins. This underscores the effectiveness of the SCDI. Therefore, determining the best-fitting distribution for the Yangtze River Basin based on individual sub-basins appears to be the most suitable approach. This novel perspective provides valuable insight for the development of a comprehensive drought index framework within the Yangtze River Basin.

The copula joint distribution method is commonly used to construct a comprehensive drought index. Because of the difficulty of multivariate copula function fitting parameters, most comprehensive drought indexes are built based on binary copula [66,67]. This paper constructed a multivariate comprehensive drought index based on the C-vine copula function, which has advantages in edge distribution fitting and joint distribution fitting. In addition, this study used AIC, BIC, and LL standards to select the joint distribution, and different sub-basins will determine the joint distribution that is more suitable for the basin according to the standard to characterize the drought situation in each area. However, the fitting of high-dimensional copula parameters is still a complex problem, and there is no research to propose a universal and accurate method to deal with multiple types of copula functions.

### 4.2. Lag Effect of the NDVI

Correlation studies have shown that the NDVI lags with climate change [68,69]. According to research by Wang et al., precipitation is an important factor affecting vegetation growth, and there is a lag effect in the response of monthly vegetation coverage to rainfall [36,70]. In the lag analysis of different regions, the lag between the NDVI and precipitation is mainly 1 to 2 months [68,71]. This temporal disparity arises because of the fact that the development of vegetation is shaped not only by prevailing climatic elements but also by the enduring impact of past weather conditions on vegetative growth. In this study, an analysis of the autocorrelation of precipitation, NDVI, and runoff revealed that the NDVI exhibits a one-month lag. By reducing the lag effect, the sensitivity of the Standardized Comprehensive Drought Index (SCDI) in capturing drought is significantly enhanced, yielding improved outcomes. Therefore, the SCDI can overcome the shortcomings of the VCI in monitoring agricultural drought lag and can capture the occurrence of drought in time. It serves as an effective evaluation metric for agricultural drought prevention and management.

The SCDI is a drought index calculated based on the NDVI. Because of the lag effect of the NDVI, it may affect the performance of the comprehensive index, so it is necessary to analyze the impact of the NDVI on the index. However, the lag of this paper was based on the long time series data analysis of the Yangtze River Basin, which may not be universal in other basins, such as the Yellow River Basin, so the NDVI lag of other basins needs to be calculated according to the vegetation coverage and time series data of each basin.

### 4.3. Synthesis of Different Types of Droughts

As a severe drought develops, various types of droughts may arise, and there is a progressive relationship in the spread of drought [15]. When drought events occur, the initial type of drought is a meteorological drought resulting from a continuous reduction in precipitation. As the drought persists, it impacts natural runoff and vegetation, leading to agricultural and hydrological drought [67]. Consequently, meteorological drought occurs rapidly, and its onset can be promptly identified. Hydrological drought generally follows a meteorological drought involving complex interactions within the underlying precipitation area. Additionally, vegetation and crops exhibit a delayed response to drought, resulting in a delayed manifestation of agricultural drought. In order to integrate the characteristics of various types of droughts, this paper constructed a comprehensive drought index.

The SCDI used in this study combines the SPI (meteorological drought), VCI (agricultural drought), and SRI (hydrological drought), offering a new method for analyzing drought in the Yangtze River Basin. Precipitation is the critical driver of drought. The SPI, represented by precipitation, can accurately capture the beginning of drought, while vegetation and runoff have a delayed effect on drought compared to precipitation. The VCI and SRI, represented by vegetation and runoff, can better capture the end of a drought. Therefore, the SCDI combines the characteristics of three elements to capture drought, and it has advantages in accurately capturing the drought process. The continuum composed of precipitation, surface cover, and soil moisture plays a vital role in the water cycle [72]. The water in vegetation and runoff will be replenished to the water cycle in the dry season. Precipitation can meet the water consumption of vegetation and runoff in the wet season,

thereby reducing the degree of drought. Therefore, the SCDI characterizes the drought degree of the entire continuum, which is more comprehensive than the univariate drought index. Drought is a complex process. In addition to precipitation, the NDVI and runoff, other influencing factors can also be considered in the future, such as river disconnection, crop coverage, and human activities, so as to construct more perfect drought indicators.

*4.4. Drought Assessment in the Yangtze River Basin*

The Yangtze River Basin is situated in humid and semi-humid regions. The process of meteorological drought evolving into agricultural and hydrological droughts, and potentially even disasters, is intricate and influenced by various factors, including precipitation, temperature, and agricultural practices [73]. As the economy and society rapidly develop and human activities increase, the interdependence between economic and social development and water resource availability in the Yangtze River Basin becomes more pronounced. Consequently, analyzing the comprehensive drought situation within the Yangtze River Basin through the perspective of an extended drought index holds significant importance. Such analysis profoundly impacts water supply security, food production, the ecological environment, and hydropower generation within the Yangtze River Basin [74].

This paper conducted a novel drought assessment of the Yangtze River Basin using the SCDI. The most severe month of drought during the study period was December 2008, during which the proportions of light, moderate, powerful, and extreme drought were 0%, 1.07%, 19.6%, and 79.33%, respectively. This finding aligns with the results of Xie et al. [75], who also reported a more severe winter drought in 2008. From the perspective of spatial feature, during all drought events, the regions both upstream and downstream of the Yangtze River are more susceptible to severe drought and extreme drought. This observation is consistent with previous studies [76], highlighting the tendency for severe drought to develop and evolve in these two regions.

These results in Figure 13 indicate a strong correlation between drought duration and intensity. Furthermore, the western region of the Yangtze River Basin demonstrated low drought frequency, high power, and long duration. On the other hand, the eastern part experienced high drought frequency, low intensity, and short duration. Moreover, drought's cumulative frequency, time, and power in the Yangtze River Basin exhibited substantial spatial heterogeneity. This observation is consistent with previous studies [72,77]. As shown in Figure 14, The areas with a high frequency of severe and extreme drought are clustered in the regions upstream of the Yangtze River, suggesting that while the frequency of drought events in the upper reaches might be low, the potential for encountering severe and extreme drought conditions remains elevated. This observation confirms the characteristic of high drought intensity in this region. These findings are consistent with the research conducted by Liu et al. [39].

**5. Conclusions**

Based on the monthly precipitation, NDVI, and runoff data from the Wuhan-Hukou Basin from 2001 to 2018, this paper constructed the SCDI using C-vine copula. The reliability and advantages of the SCDI in drought monitoring were comprehensively verified through assessments of its consistency, sensitivity, accuracy, false negative rate, false positive rate, and specific historical drought events. Furthermore, this paper analyzed the applicability of the SCDI in the Yangtze River Basin and explored the spatiotemporal evolution characteristics of drought in the region. The conclusions are as follows:

(1)  The SCDI exhibits a superior drought monitoring performance. It accurately identified the onset and cessation of drought, characterized all drought events in the Wuhan-Hukou Basin, and maintained low false negative and false positive rates. By combining precipitation, NDVI, and runoff, the SCDI simultaneously characterizes meteorological, hydrological, and agricultural droughts.

(2)  The NDVI exhibits a lag effect on the construction of the SCDI. The sensitivity of the SCDI to the VCI increased from 47.8% to 53% when shifting the NDVI time series

data forward by one month in sequence during the SCDI's calculation. This indicates that the SCDI demonstrates a more effective drought monitoring capability after accounting for the lag effect of the NDVI.

(3) The SCDI applies to all third-level sub-basins of the Yangtze River Basin. While there is some spatial heterogeneity in correlation, consistency, and sensitivity, the quantitative indicators' results fell within a reasonable range, and the false negative and false positive rates remained between 0 and 20%. Therefore, the SCDI exhibits good applicability in the Yangtze River Basin.

(4) From the perspective of time, the droughts of all grades in the Yangtze River Basin showed a fluctuating trend from 2001 to 2018, with December 2008 being the most severe drought month. From the perspective of space, the characteristics of drought from 2001 to 2018 exhibited evident spatial heterogeneity. The western region experienced low drought frequency, high intensity, and long duration, while the eastern part showed the opposite.

**Author Contributions:** H.W., W.Z., X.L., C.J. and S.H. were involved in the intellectual elements of this paper; H.W., W.Z., X.L. and C.J. designed the research; H.W. conducted the research and wrote the manuscript; C.J. and S.H. helped with the data arrangement and analysis. Conceptualization, H.W. and W.Z.; Formal analysis, H.W. and C.J.; Funding acquisition, X.L.; Investigation, and S.H.; Methodology, H.W. and W.Z.; Project administration, X.L.; Resources, H.W.; Software, H.W. and C.J.; Supervision, W.H.; Visualization, W.H.; Writing—original draft, H.W.; Writing—review and editing, H.W. and W.Z. All authors have read and agreed to the published version of the manuscript.

**Funding:** This research was funded by the Key R&D Program of Hubei (2022BCA080) and the National Natural Science Foundation of China (Grant No. 41501584).

**Data Availability Statement:** Data are available upon request.

**Conflicts of Interest:** The authors declare no conflict of interest.

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
