# Peer review of "Copula-Based Joint Drought Index Using Precipitation, NDVI, and Runoff and Its Application in the Yangtze River Basin, China"

_remotesensing, doi:10.3390/rs15184484_

Round 1

Reviewer 1 Report

This article proposes a new comprehensive drought index and assesses its applicability in the Yangtze River Basin. The findings of this paper can assist us in monitoring droughts more accurately and effectively, providing valuable insights for drought prevention and preparedness. However, I believe that before its acceptance, a revision should be conducted. Below are detailed comments for the manuscript:

1. Line 108-109: The author should add references or supporting materials to substantiate the statement “Over the last few decades, the Yangtze River Basin has encountered numerous extreme drought events as a consequence of both changes in climate patterns and human interventions."

2. In the data section of chapter 2.2 of the manuscript, the author should supplement the detailed information such as the name, time and spatial resolution of the dataset.

3. In Figure 2, the format of the word "distributions" in "Fitting marginal distributions" is incorrect. In addition, AIC is wrongly written as ACI. The pattern can be improved to make it more beautiful.

4. Line 158: The title of section 2.3.1, "single element drought indices calculation," should have a capital letter at the beginning to maintain consistency with other titles.

5. SPEI has been widely applied in drought monitoring and assessment under the backdrop of global warming. The author may consider comparing and discussing both the SPEI and the SCDI.

I think the quality of English Language is good, but the authors should pay attentions to checking for minor spelling errors.

Reviewer 2 Report

General comment. The article needs to strengthen the physical background regarding the characteristics and applicability of the developed index.

1.  (2.3.4. Characteristics of drought)

Although there is currently no clear and generally accepted criterion for drought, the drought process is separated from the rainless period. Usually, meteorological drought is fixed on a time scale from 1 month, smaller time scales are referred as rainless periods. How legitimate is it to assume that there are two droughts in one month? What physical mechanism is consistent with criterion 3 introduced by you?

2. As indicated  in the article, the selected three parameters of precipitation, NDVI and runoff characterize three types of drought: meteorological, agrometeorological and hydrological, respectively. The time scales of these types of droughts are different, and although they usually interact with each other, it is not always one type of drought that triggers another. How does the index SCDI take into account the diversity of time scales and the interdependence of drought types? What specific type of drought does the proposed index assess, and on what time scale?

3. (3.4.1. and Fig.11). Based on Fig. 11, it can be concluded that droughts are observed every year in the Yangtze River basin, which is obviously related to the climatic regime of precipitation and temperature. However, in this case, is it correct to call the considered processes as a drought for the entire basin, or is it just interannual and intraannual variability?

4. The results presented in 3.3 raise doubts about the universality and expediency of using the SCDI index for all regions due to the low consistency of its components, which, obviously, do not take into account the characteristics of local places.

The text is desirable to show a native speaker. 

Reviewer 3 Report

This study proposes the standardized comprehensive drought index (SCDI) using a novel approach that utilizes the joint distribution of C-vine Copula0. And analyzes the reliability and effectiveness of SCDI in detecting drought events through quantitative indicators and assesses its applicability in the Yangtze River Basin. However, there are still some problems in the paper, as follows:

1.L69,First ? Why not Firstly?

2.Please add additional references for the introduced concepts in particular for Line 35-68 but generally for all the section if possible.

3.L83-95, please try to rewrite and express the purpose of this study and the problems that need to be solved, which makes me confused and don 't know what your research doing.

4.L71-72, NDVI should indicate the full name when it appears for the first time, and other abbreviations are the same.

5.Fig2. What means ACI?

6.L154-155, what is the grading threshold? How to determine? Where is it reflected?

7.Section2.3.1, SPI includes 1, 3, 6, 12, and 24 months scale, why choose 1 month scale ? The calculation data of SPI also includes temperature data. How is this kind of data obtained ? Which data source is used? In addition, how is SPI calculated? Using station data or interpolated raster data ? Based on what kind of programming calculation ?

8.L290-305, I think this paragraph is not the result, should be adjusted to the method. Similar explanations in the results should be put into the method. Please re-comb the relevant explanations in the results. The results part only presents the results of this study and simple analysis. The correlation analysis can be put into the discussion, such as L332-340, 341-344, L394-396, L559-567, L604-614.

9.Section3.2, the effective value of VCI is not between [-1,1]? Please confirm your VCI result.

10.L172-178 is mainly to propose VCI, why use a large space to introduce NDVI?

11.L123,L126 added (accessed on X XX 202X) after URLs.

12.L123, please confirm whether there is the expression of average precipitation ? The general precipitation is expressed as cumulative value.

13. Section2.2 Please describe the spatio-temporal resolution of all data in detail and how to unify the spatio-temporal resolution.

14.3.4.1 and 3.4.2 consistent title ? In addition, how to determine the lag time ?

15.I suggested to reorganize the discussion part and rewrite the discussion. Combined with the main results and conclusions, the detailed analysis could combined with the main results. The discussion is written in chapters, and the similarities and differences between similar research and this study are discussed. The main reasons for the similarities and differences, and indicate the limitations of this study and the focus of subsequent research.

On the whole, I think the language of the paper is not concise and standardized enough, and still needs further polishing.

Round 2

Reviewer 2 Report

Table 1. 

The grade of drought cannot be normal. In this interpretation, the reader can understand that this is some kind of normal manifestation of drought. But in fact, this criterion means that, for example, according to SPI, precipitation was close to normal, there is no precipitation anomaly (or not anomalies of other characteristics for other indices). It would be more correct to call this grade as Neutral, which means the absence of significant deviations of the index from the normal (climatic) state.

Reviewer 3 Report

1.ACI does not seem to be replaced in Figure 2 ? 

2.You have explained the normalization of the four indicators, which means that the values of all indicators are within a certain range. If you make the effective value range of the four indicators between [− 2.5 ,2.5],  How did you deal with it ? I hope you will explain. The specific calculation method and formula ? 

I have no opinion on other revisions, and can reach the level of publication after interpretation.
